# Applications of Photonics in Agriculture Sector: A Review

**DOI:** 10.3390/molecules24102025

**Published:** 2019-05-27

**Authors:** Jin Yeong Tan, Pin Jern Ker, K. Y. Lau, M. A. Hannan, Shirley Gee Hoon Tang

**Affiliations:** 1Institute of Power Engineering, College of Engineering, Universiti Tenaga Nasional, Kajang 43000, Selangor, Malaysia; adrian_tan_jy@hotmail.com (J.Y.T.); kylau@uniten.edu.my (K.Y.L.); Hannan@uniten.edu.my (M.A.H.); 2Microbiology Unit, Department of Pre-clinical, International Medical School, Management and Science University, University Drive, Off Persiaran Olahraga, Seksyen 13, Shah Alam 40100, Selangor, Malaysia; shirley_tang@msu.edu.my

**Keywords:** agriculture, photonics, imaging, spectral imaging, spectroscopy

## Abstract

The agricultural industry has made a tremendous contribution to the foundations of civilization. Basic essentials such as food, beverages, clothes and domestic materials are enriched by the agricultural industry. However, the traditional method in agriculture cultivation is labor-intensive and inadequate to meet the accelerating nature of human demands. This scenario raises the need to explore state-of-the-art crop cultivation and harvesting technologies. In this regard, optics and photonics technologies have proven to be effective solutions. This paper aims to present a comprehensive review of three photonic techniques, namely imaging, spectroscopy and spectral imaging, in a comparative manner for agriculture applications. Essentially, the spectral imaging technique is a robust solution which combines the benefits of both imaging and spectroscopy but faces the risk of underutilization. This review also comprehends the practicality of all three techniques by presenting existing examples in agricultural applications. Furthermore, the potential of these techniques is reviewed and critiqued by looking into agricultural activities involving palm oil, rubber, and agro-food crops. All the possible issues and challenges in implementing the photonic techniques in agriculture are given prominence with a few selective recommendations. The highlighted insights in this review will hopefully lead to an increased effort in the development of photonics applications for the future agricultural industry.

## 1. Introduction

Light constitutes a collection of particles known as photons, propagated in the form of waves [1]. In physics, light often relates to radiation in the entire electromagnetic spectrum, encompassing X-rays, ultraviolet, visible light, infrared, and microwaves among others [2]. The unique electromagnetic properties of light have intrigued academics across the globe and the earliest study can be traced back to the early 17th century [3]. As time passes, the accumulation of knowledge and technological advancement have gradually shaped the canvas for light-related research, leading to the establishment of the field of optics and photonics.

Optics can be defined as a branch of physics that studies the behavior and properties of light as well as the interaction of light with other matter [2]. Meanwhile, photonics can be regarded as the application of light through the systematic generation, control and detection of photons [2,4]. Despite the distinction between optics and photonics, both terminologies have often been used interchangeably in the literature to collectively represent the science and application of light [1].

Optics and photonics have influenced various engineering applications, transforming the landscape of various fields and improving the lives of mankind. One of the main applications of optics and photonics can be seen in the field of communications. Knowledge of optics and photonics has been used to develop optical fibers which help to cater for the needs of broadband Internet service in this “data hungry” era. Furthermore, optics and photonics have been used in the manufacturing of modern displays such as liquid crystal display (LCD), organic light-emitting diode (OLED), flexible display and such. Solar cells for energy harnessing too illustrate another application of optics and photonics. Not least, optics and photonics have also been applied in more sophisticated areas such as security surveillance, medical imaging, quantum computing and more [1].

Amidst the modern and complex solutions discussed earlier, it often slipped our minds that optics and photonics can be readily integrated into the field of agriculture. The simplest examples would be the adjustment of plantation direction for optimum sunlight exposure, as well as the usage of incandescent light bulbs in egg incubation and hatching [5]. Over recent decades, academics have been alerted to the potential of optics and photonics in the agricultural industry. This has led to progressive developments that utilize optics and photonic techniques in maximizing the quality and productivity of agricultural products.

This paper aims to review some of the most popular optics and photonic techniques in agriculture, namely imaging, spectroscopy and spectral imaging. In addition, existing applications of each technique in the agricultural industry will also be compiled. A comprehensive discussion will also be made to gauge the potential of exploiting optics and photonic techniques in the agricultural sector with the intention of improving the quality and productivity of the agricultural products at a reduced labor cost.

## 2. Classification of Photonics Systems in Agriculture

Quantity and quality have always been the primary foci in the field of agriculture. The governing of these attributes is anticipated to be more crucial in the upcoming years. This prediction is based on the constant increase in global population as well as heightened expectations for healthy food sources. However, the agricultural field faces great pressure under globalization. The transformation of the global economic landscape makes agricultural activities seem less profitable in contrast to other industrial activities. The outflow of the workforce makes it increasingly expensive and difficult to meet the demands of agricultural activities.

As a result, modern technology has been integrated into the agricultural field to maximize output efficiency at minimum labor force. Similar to other industries, automation systems have been applied in stages of agricultural activities to reduce a dependency on manual labor [6]. These systems require optics and photonics techniques to complement them, providing the required ‘sight’ for operations. These vision requirements have been fulfilled by optics and photonic techniques such as imaging, and spectral imaging. These techniques provide machine vision at high dynamic range, high resolution and high accuracy in a non-destructive, non-contact and robust manner [5]. In the subsections below, details of ESS configurations, their classifications and structures have been illustrated.

### 2.1. Imaging Technique

The imaging technique is analogous to the function of the human eye. It captures the image of the subject for necessary calculations and measurements before performing the final evaluations [7]. The imaging technique is essential for collecting spatial, color [6] and even thermal [8] information of the subject of interest. Therefore, imaging techniques are typically operated in an active manner. The active imaging technique involves image acquisition under two major light sources, namely visible light and infrared sources. Images under visible light can be easily acquired with any standard camera modules. On the other hand, images under exposure to infrared can be acquired with special infrared camera modules [8].

Image acquisition under visible light is similar to our daily photography. The image acquisition process under this light source is straightforward and images captured are usually rich in details and colors. However, complexity often arises while performing analysis on these images due to illumination variations. For instance, images captured outdoors vary under sunny and cloudy conditions. Meanwhile, images captured indoors is categorized by natural light, incandescent and fluorescent conditions [7].

The acquired image will then undergo pre-processing to convert it into an appropriate format before further analysis. Pre-processing tasks may include exposure correction, color balancing, noise reduction, sharpness increase or orientation change. Next, the process of feature detection and matching as well as segmentation is performed on the pre-processed image to extract the object or region of interest. Finally, the subject of interest is analyzed with proper analysis algorithms in the respective area of application [9].

The imaging technique can be easily applied in the simple analysis of static-positioned objects or even in more complex areas which involve moving targets, such as visual navigation and behavioral surveillance. These achievements were made possible by utilizing the spatial information acquired through the imaging technique for position triangulation and motion guidance [7,9]. In image processing, the computer imaging technique has been employed to create, edit, and display graphical images, characters, and objects. The computer image analysis technique is a broad field which consists of computer domains and applications in food quality evaluation [10,11], grading and the sorting of agricultural products [12,13], as well as harvesting the crops [14], and estimating moisture content in the drying stage for the storability of the food product [15]. Computer imaging contributes to the development of digital agriculture. For instance, weed detection and fruit grading systems with digital imaging techniques are cost effective systems in achieving ecological and economically sustainable agriculture [16]. 

### 2.2. Spectroscopy Technique

In contrast to the imaging technique, the spectroscopy technique enables the ‘sight’ of properties that are invisible to the naked eye. The spectroscopy technique functions by extracting spectral information from the sample of interest. The spectral information is obtained when light interacts with the composition of the sample. This interaction leads to changes in the intensity or frequency and wavelength of the initial light source, ultimately defining a spectrum which acts as the fingerprint of the sample [17].

Similar to the imaging technique, variations do exist for spectroscopy. These variations are categorized by the nature of interaction between the light source and the sample when the spectroscopy measurement is conducted. In the agricultural field, the commonly adopted spectroscopy techniques are ultraviolet-visible (UV-VIS) spectroscopy, fluorescence spectroscopy, infrared (IR) spectroscopy, and Raman spectroscopy [17].

#### 2.2.1. Ultraviolet-Visible (UV-VIS) Spectroscopy

The ultraviolet-visible (UV-VIS) spectroscopy is conducted in both the ultraviolet (UV) and visible light (VIS) band, spanning wavelengths from 100 nm to 380 nm (UV) and from 380 nm to 750 nm (VIS). The principle governing the UV-VIS spectroscopy is Beer-Lambert’s law, which is expressed by (1) and (2):(1)$$I=\;I_{0}10^{-\varepsilon cl},$$
(2)$$\ln\frac{I_{0}}{I}=\ln\frac{1}{T}=\;\varepsilon cl=A$$
where *I*_0_ and *I* are intensity of light entering and leaving a sample respectively, *ε* is the extinction molar coefficient, *c* is the molar concentration of substance, *l* is the thickness of sample (cm), *T* is transmittance and *A* is absorbance [18].

A typical model that illustrates Beer-Lambert’s law can be seen in Figure 1. It can be observed that as light propagates through a sample, a portion of the incidental light source will be absorbed by the molecules in the sample, while the remaining light rays will transmit and escape across the sample. The ratio between the intensity of the incident and escaped rays defines the absorbance of light by the sample. This value of light absorbance is of main interest in UV-VIS spectroscopy. As in Equation (2), light absorbance is dependent on *ε*, *c*, and *l* [18]. The absorbance value(s) at a single or multiple wavelength(s) will then be used to measure the concentration of compounds in a sample [19,20,21,22,23].

#### 2.2.2. Fluorescence Spectroscopy

Fluorescence spectroscopy is distinct from other spectroscopy techniques in terms of the emission of light when incident rays from an ultraviolet or visible light source is absorbed by fluorescent molecules present in a sample. These fluorescent molecules are known as fluorophores and commonly known examples include quinine, fluorescein, acridine orange, rhodamine B and pyridine 1 [24].

The fluorescence phenomenon can be explained with a Jablonski diagram illustrated in Figure 2. It should first be understood that fluorescence involves the three electronic states of a fluorophore molecule, namely the singlet ground, first and second electronic states. These states are represented by S_0_, S_1_ and S_2_ in Figure 2. The key condition for fluorescence to occur is the excitation of the molecule from the ground state, S_0_ to either electronic states S_1_ or S_2_ upon the absorption of light. If the molecule reaches the S_2_ state, internal conversion or vibrational relaxation will occur, returning the molecule to the lower S_1_ state without radiation emitted. From here, the molecule will again return to the S_0_ while emitting light which has equal energy as the energy difference between S_0_ and S_1_. This light emission is known as fluorescence and this condition typically occurs 10^-8^ seconds after the initial excitation [17].

Fluorescence spectroscopy is highly specific and highly sensitive. The high specificity of the technique arises from the usage of both the excitation and emission spectra; whereas high sensitivity is achieved as radiation measurements are made against absolute darkness. These characteristics however limit the independent usage of the technique [17]. As a result, fluorescence spectroscopy is often combined with high performance liquid chromatography (HPLC) [25]. Variations may also be implemented in the excitation and emission wavelengths, forming the synchronous fluorescence spectroscopy (SFS) [26].

#### 2.2.3. Infrared (IR) Spectroscopy

Infrared (IR) spectroscopy operates within the IR band with wavelengths from 780 nm to 1 mm. The IR band can be further broken down into three sub-bands, namely near-infrared (NIR; 780 nm to 5 µm), mid-infrared (MIR; 5 µm to 30 µm) and far-infrared (FIR; 30 µm to 1 mm). In agriculture-related optics and photonics, the NIR and MIR bands are of greater interest [17].

IR spectroscopy obtains the spectral information of a subject due to molecular vibrations under the excitation of an IR light source. In general, molecular vibrations occur when there exist normal modes of vibrations. A normal mode of vibration (or fundamental) refers to the phenomenon in which every atom in a molecule experiences a simple harmonic oscillation about its equilibrium position. These atoms oscillate in phase at the same frequency while the center of gravity of the molecule remains unchanged. A typical molecule has 3N-6 fundamentals (3N-5 for linear molecules), where N refers to the number of atoms. The diatomic molecular vibrations are illustrated in Figure 3 [27].

Molecular vibrations, which occur regardless the presence of IR light source, result in an increase in light absorption. These peaks in absorption form specific bands in the IR spectrum that correspond to the specific frequencies in which molecular vibrations occur. This allows the easy identification of the molecular structure in a sample since different molecules have different vibration frequencies [27]. This unique frequency ‘fingerprint’ is exceptionally beneficial in the analysis of complex molecules that contains functional groups such as –OH, –NH_2_, –CH_3_, C=O, C_6_H_5_– and more. For instance, the C_6_H_5_– group forms peaks at wavenumbers from 1600 cm^−1^ to 1500 cm^−1^ (wavelengths from 6.25 µm to 6.67 µm) whereas the C=O group exhibits high absorption at wavenumbers from 1800 cm^−1^ to 1650 cm^−1^ (wavelengths from 5.56 µm to 6.06 µm) [28].

#### 2.2.4. Near-Infrared (NIR) Spectroscopy

The near-infrared (NIR) spectroscopy operates within the NIR band with wavelengths from 780 nm to 5 µm. The absorptions within the NIR band exist due to overtones and combinations of the fundamental vibrations. Overtones refer to higher frequencies that are multiples of the fundamental frequency. Meanwhile, combinations involve interactions between two or more vibrations occurring simultaneously, resulting in a frequency which is the sum of multiples of the respective frequencies. A majority of the absorptions in the NIR band are due to vibrations of the C–H, O–H and N–H bands. The S–H and C=O bonds too potentially contribute to these absorptions. Several assignments of the NIR absorption bands can be seen in Table 1 [29].

NIR spectroscopy, which is a non-destructive measurement, enables the simultaneous identification of components in a single sample within a short period of time, making it a preferable replacement for various chemical techniques. However, consideration should be taken into account as this technique requires initial calibration with samples of known composition, requiring great expenses of time and resources. Not least, frequent recalibration and issue of instrument interoperability might affect the practicality of the NIR spectroscopy technique [29].

#### 2.2.5. Mid-Infrared (MIR) Spectroscopy

The mid-infrared (MIR) spectroscopy operates within the MIR band with wavelengths from 5 µm to 30 µm (wavenumbers from 4000 cm^−1^ to 400 cm^−1^; note the presence of slight overlapping with NIR). The absorptions that occur within the MIR band are due to fundamental vibrations and can be segregated into four regions, namely the X–H stretching region (4000 cm^−1^ to 2500 cm^−1^), triple-bond region (2500 cm^−1^ to 2000 cm^−1^), double-bond region (2000 cm^−1^ to 1500 cm^−1^) as well as the fingerprint region (1500 cm^−1^ to 600 cm^−1^) [27].

The X–H stretching region is due to vibrations from O–H, C–H and N–H stretching. The triple-bond region arises from vibrations of C≡C and C≡N bonds. Besides, the double-bond region relates to C=C, C=O and C=N vibrations. Lastly, the fingerprint region roots on bending and skeletal vibrations. Table 2 lists some of the common examples of MIR absorption bands [27].

MIR spectroscopy is effective since it provides information on structure-function relationships while performing quantitative analysis. The structure-function relationships are useful in food research and quality control, making MIR spectroscopy a crucial technique in the field of agriculture. The Fourier transform process is often bundled with MIR spectroscopy for data analysis, forming the popular Fourier transform infrared spectroscopy (FTIR) technique [27].

#### 2.2.6. Raman Spectroscopy

Raman spectroscopy (RS), similar to IR spectroscopy, is another form of vibrational spectroscopy technique. RS obtains the spectral information of samples due to the occurrence of Raman effects [30]. Prior to understanding the Raman effects, one should look into the light scattering schemes that occur when incident photons interact with molecules in the sample. The possible light scattering schemes are illustrated in Figure 4. In the case of elastic scattering or Rayleigh scattering, the excited photons experience no change in energy content upon returning to ground state. Alternately, in the case of inelastic scattering or Raman scattering, the excited photons may lose (Stokes’ shift) or gain (Anti-Stokes’ shift) energy equivalent to the vibrational energy changes in the atoms of the molecules. This affects the motion of the atoms as well as the polarizability of the molecule. The change in molecule polarizability results in increased Raman intensity, ultimately forming the Raman spectrum when plotted across the investigated wavenumbers. However, this effect is weak as the probability of energy exchange is low [30].

The RS technique is gaining popularity as it enables the identification of molecular structure through the characteristic wavenumber in which vibrations occur. Furthermore, samples can be studied in the absence of a solvent as water causes weak Raman scattering. Not least, this technique is instantaneous and may undergo intensity enhancement. However, this technique is not without limitations. Due to the low probability of Raman scattering, this technique requires high concentration of samples. Moreover, sample molecules may experience photo degradation due to excitation of electronic absorption bands. The existence of fluorescence from impurities may disrupt the results obtained as well. These limitations aside, the RS technique can be combined with IR spectroscopy to deliver satisfactory results as summarized in Table 3 [30].

#### 2.2.7. Additional Spectroscopy Techniques

Apart from the popular spectroscopy techniques discussed earlier, existing studies presented additional variations of spectroscopy techniques which may be more complex in nature. For instance, dielectric spectroscopy has been utilized in agricultural inspections. Dielectric spectroscopy involves the inspection of dielectric properties or permittivity of samples over broad frequency ranges. Dielectric properties or permittivity refers to the ability of samples to store electrical energy in the electric field. In this spectroscopy technique, the permittivity is a complex permittivity relative to the free space, and this complex number is represented by (3):(3)$$\varepsilon =\varepsilon^{\prime }+j\varepsilon^{\prime \prime }$$
where the real part, *ε′*, is the dielectric constant and the imaginary part, *ε″*, is the dielectric loss factor which covers losses due to dipolar relaxation and ionic conduction [31]. Another spectroscopy variation is the nuclear magnetic resonance (NMR) spectroscopy technique. The NMR spectroscopy gains spectral information of samples from the interaction between the magnetic moments of nuclei of various atoms and the applied magnetic fields. The two common phenomena that give rise to the NMR spectra are chemical shift and J-coupling [32].

A chemical shift occurs due to different resonant frequencies present in nuclei of the same species. The difference in resonant frequencies is a result of shielding effect from electrons surrounding the nuclei. The shielding effect is sensitive to chemical environments, hence allowing the characteristic identification of specific molecular functional groups [32].

The J-coupling phenomenon is also known as indirect (scalar) spin-spin coupling. This coupling effect results in splitting of spectroscopic lines into multiplets. The J-coupling occurs between two nuclei or groups of nuclei and is governed by the polarization of electrons on the chemical bonds connecting these nuclei. The polarization scheme is in turn dependent on the instant orientation of the nuclear magnetic moments in the presence of a magnetic field [32].

#### 2.2.8. Spectroscopy Processing and Analysis

The raw spectral data undergoes pre-processing or pre-treatment in order to reduce noise and correct baseline variations. The common pre-treatment techniques are multiplicative scattering correction (MSC), standard normal variate (SNV), Savitzky-Golay smoothing as well as first and second derivatives [33,34].

Upon the completion of pre-processing or pre-treatment, the data set undergoes multivariate analysis to select and extract wavelengths that contain useful information. This aids in rectifying issues of collinearity, band overlapping and interaction between spectral variables. The results from multivariate analysis will be used to develop calibration models for calibration and prediction purposes [33,34].

The developed calibration models can be categorized according to the nature of the utilized multivariate analysis such as linear regression or nonlinear regression. Calibration models based on linear regression are built from partial least squares (PLS), interval partial least squares (iPLS), synergy interval partial least squares (SiPLS) or successive projections algorithm (SPA). Meanwhile, calibration models based on nonlinear regression are constructed from principal component analysis (PCA), independent component analysis (ICA), support vector machines (SVM), artificial neural networks (ANN) or a genetic algorithm (GA) [33,34].

The robustness of the final calibration model is evaluated from its ability to perform calibration and prediction. The calibration performance of the model is determined from the root mean square error of calibration (RMSEC) and the correlation coefficient (R_C_) in the calibration set. Meanwhile, the prediction performance of the model is identified from the root mean square error of prediction (RMSEP) and the correlation coefficient (R_P_) in the prediction set. Ideally, an effective model should register low RMSEC and RMSEP, with minimum difference between RMSEC and RMSEP. Not least, higher R_C_ and R_P_ are preferable [33,34].

### 2.3. Spectral Imaging Technique

The spectral imaging technique is a combination of both imaging and spectroscopy techniques discussed earlier. Being a combinational technique, the spectral imaging technique preserves the best of both worlds, allowing the simultaneous extraction of spatial and spectral information from the inspected sample [35,36].

#### 2.3.1. Classes of Spectral Imaging

The spectral imaging technique acquires multiple images of the same subject at varying wavelengths. The resulting spectral images are three-dimensional (3-D) in nature, consisting of two spatial dimensions (row, x, and column, y) and one spectral dimension (wavelength, λ). Variations of spectral imaging technique are determined by the continuity of data in the wavelength dimension, branching out into hyperspectral imaging and multispectral imaging [35,36].

In general, hyperspectral imaging obtains spectral images in continuous wavelengths, whereas multispectral imaging registers spectral images at discrete wavelengths. Hyperspectral imaging acquires large number of images at high spatial and spectral resolutions. Due to the high volume of data, hyperspectral imaging requires long image acquisition time and involves complex algorithms for image analysis. Despite the complexity, hyperspectral imaging is essential for fundamental research and is the basis for multispectral imaging [35,36].

Multispectral imaging acquires spectral images at a significantly smaller number compared to hyperspectral imaging. Spectral images will only be acquired at optimal wavelengths predetermined from the analysis of dataset obtained through hyperspectral imaging. A smaller number of interested wavelengths allows rapid image acquisition and requires simpler image analysis algorithms. This characteristic of optimum data volume makes multispectral imaging perfectly suited for real-time in-field applications [35,36].

#### 2.3.2. Spectral Image Acquisition Methods

There are several methods in which spectral imaging systems acquire spectral images. The methods are point scan, line scan and area scan as illustrated in Figure 5 [35]. The point scan (whiskbroom) method acquires the spectrum of a single pixel in each scan. A complete hyperspectral cube will be generated as the detector moves from pixel to pixel along the two spatial axes (x and y). The point scan method is similar to a normal spectroscopic approach. Since it cannot cover a large sample area, the point scan method is time consuming and unsuitable for fast image acquisition [35,36].

The line scan (pushbroom) method, in each scan, acquires a slit (line) of spatial information together with the spectrum of every pixel along the line. A complete hyperspectral cube will be formed when scans are repeated along the direction of motion (*x*). The operation characteristic of the line scan method makes it suitable to acquire spectral images of moving samples. Hence, this method is usually combined with conveyor belt systems, making it a popular method in practical production lines. However, the exposure time should be short and accurately selected to allow uniform exposure at all wavelengths [35,36].

The area scan (band sequential) method, on the other hand, acquires a 2-D grayscale image comprising of complete spatial information in a single wavelength. A complete hyperspectral cube is generated through image stacking when scans are performed along the spectral axis (λ). The nature of the area scan method makes it more suited for the imaging of stationary samples instead of moving samples. In short, among the image acquisition methods discussed, line scan and area scan are greatly preferred over point scan for both hyperspectral and multispectral imaging on the basis of time consumption [35,36].

#### 2.3.3. Spectral Imaging Sensing Modes

Spectral imaging may have varying sensing modes as illustrated in Figure 6. The sensing modes are determined by the positions of the light source and the detector, forming variations such as reflectance, transmittance and interactance modes. In reflectance mode, the detector collects the light reflected off the illuminated surface. This sensing mode is suitable for identifying external features of samples such as size, shape, color, texture and defects. However, when selecting this mode, the detector should be properly positioned to avoid specular reflection [36].

The transmittance mode operates by having the detector collect light rays transmitted through inspected samples. In this sensing mode, the light source and the detector will be placed in opposite direction to each other. Due to the absorption of light rays in a sample, the detected signal will be relatively weak and dependent on sample thickness. Hence, the transmittance mode is commonly applied in the internal inspection of relatively transparent samples [36].

Meanwhile, the interactance mode overcomes the limitations of both the reflectance and transmittance modes. This sensing mode exhibits less surface effect compared to reflectance mode. At the same time, it allows detection in deeper layers of a sample without being affected by sample thickness as in transmittance mode. This advantageous sensing mode is set up by installing the light source and the detector at the same side and parallel to each other [36].

#### 2.3.4. Spectral Imaging System Construction

The variations in spectral imaging lead to a diversity of instruments during the construction of a spectral imaging system. In general, a spectral imaging system is made up of a light source, a wavelength dispersive device and an area detector [35,36].

The light source for a spectral imaging system can be classified into illumination and excitation sources. Illumination light source is selected when measurements involve changes in the intensity of the incident rays upon light-sample interaction. The spectral composition of the incident source will not experience any changes. Such interaction is commonly observed in reflectance and transmittance sensing modes. Broadband lights are normally used as illumination sources. An example of illumination light source is the quartz tungsten halogen (QTH) lamp which is capable of generating a smooth spectrum in the visible to infrared range. Besides, the broadband light emitting diode (LED) has gained popularity over time due to its low power consumption, low heat generation, small size and long lifetime [35,36].

Excitation light source is usually selected when measurements involve changes in the frequency and wavelength of the incident rays. Interactions of this nature usually involve fluorescence phenomenon or Raman scattering effect. Narrowband lights are frequently used as excitation sources. A popular excitation light source is the laser which generates powerful monochromatic rays. Not least, UV fluorescent lamp, narrowband LED, high-pressure arc lamp (xeon arc lamp) and low-pressure vapor lamp (mercury vapor lamp) add to the family of excitation light sources [30,31].

The core component of the spectral imaging system is the wavelength dispersive device. A wavelength dispersive device disperses broadband light into different wavelengths to be projected to the area detector. Examples of wavelength dispersive devices include the imaging spectrograph, electronically tunable filter and beam splitting device [35,36].

Compared to traditional spectrograph, an imaging spectrograph extracts both spatial and spectral information. The imaging spectrograph disperses the broadband light illuminated onto different spatial areas of a sample into different wavelengths. This is achieved through diffraction gratings. The two most popular imaging spectrographs are the prism-grating-prism (PGP) imaging spectrograph which uses transmission diffraction gratings and the Offner imaging spectrograph that uses reflection diffraction gratings [35,36]. These variations of imaging spectrographs are commonly applied in line scan acquisitions [37].

An electronically tunable filter utilizes electronic devices to extract the required wavelength. Current electronically tunable filters can be categorized into the acousto-optic tunable filter (AOTF) and liquid crystal tunable filter (LCTF). An AOTF utilizes an acoustic transducer to generate high frequency acoustic waves that change the refractive index of a crystal. The crystal with varied refractive index will only allow the passage of light rays at the specified wavelength. Meanwhile, a LCTF transmits light at the required wavelength through electronically controlled liquid crystal cells [35,36]. These electronically tunable filters allow fast and flexible wavelength switching compared to mechanical filter wheels. They too exhibit advantages of high optical throughput, narrow bandwidth and broad spectral range [38].

Unlike the electronically tunable filter, a beam splitting device allows spectral images to be obtained simultaneously at multiple wavelengths. The beam splitting device divides light into several parts and passes them through bandpass filters which correspond to the required wavelengths. The beam splitting device can be categorized into color splitting and neutral splitting. In color splitting, light rays at a particular waveband are directed to each output, whereas, in neutral splitting, an equal portion of the total light energy is directed to each output [35]. The multiple wavelength acquisition characteristic makes the beam splitting device suitable to be installed in multispectral imaging systems [39].

A spectral imaging system will not be complete without an area detector. The area detector is responsible for collecting light rays which will eventually form the spectral images of the inspected sample. The common categories of area detector are the charge-couple device (CCD) camera and the complementary metal-oxide-semiconductor (CMOS) camera [35,36].

A CCD camera is made up of millions of photodiodes (pixels) that are closely arranged to form an array. These light sensitive photodiodes convert the incident photons into electric charges that correspond to the intensity of the exposed incident rays. The accumulated electric charges at each photodiode will then be moved out of the array to be quantified for spectral image formation [35,36]. One of the common CCD cameras is the silicon CCD camera. The silicon CCD camera exploits the sensitivity of silicon under visible light to perform image acquisition in visible and short-wavelength near-infrared bands [40]. Indium gallium arsenide (InGaAs) CCD camera is another CCD camera variation constructed from InGaAs, an alloy between indium arsenide (InAs) and gallium arsenide (GaAs) which is sensitive in the near-infrared band [41]. Not least, mercury cadmium telluride (MCT or HgCdTe) CCD camera built from HgCdTe, an alloy between mercury telluride (HgTe) and cadmium telluride (CdTe), enables sensing in the long-wavelength near-infrared and mid-infrared band [42].

Comparatively, the CMOS camera is similar to the CCD camera by having a collection of photodiodes (pixels) that convert light rays into electrical charges. The difference, however, lies in the quantification process of the electric charges. Opposed to the remote quantification in the CCD camera, the CMOS camera allows electric charges at each pixel to be independently and instantaneously read by the transistor attached to each photodiode [43]. This unique characteristic allows the CMOS camera to compete with the CCD camera in terms of high imaging acquisition speed, blooming immunity, low cost, low power consumption and small size. However, careful note should be taken as the CMOS camera is susceptible to noise due to on-chip signal transmissions, resulting in lower sensitivity and dynamic range when pitted against CCD camera [36].

#### 2.3.5. Spectral Imaging Processing and Analysis

The raw spectral image data obtained via the spectral imaging technique comes in different formats according to the image acquisition method used. The common formats are Band interleaved by pixel (BIP), band interleaved by line (BIL) and band sequential (BSQ). The BIP format results from the point scan method and stores the complete spectrum of each pixel sequentially. The BIL format comes with the line scan method and stores the complete spectrum of each line in order. Lastly, the BSQ format relates to the area scan method and stacks the spatial image continuously obtained at each wavelength [35,36].

Similar to the imaging and spectroscopy techniques, the raw spectral image data in BIP, BIL and BSQ formats should undergo pre-processing in both the spatial and spectral aspects before being utilized for further analysis. The raw spectral image, which represents detector signal intensity, will first undergo flat-field calibration or reflectance calibration to form useful reflectance or absorbance image. From the spatial aspect, the generated reflectance image can be further improved through image enhancement processes such as edge and contrast enhancement, magnifying, pseudo-coloring and sharpening. Noise reduction can also be achieved through spatial filtering, Fourier transform (FT) and wavelet transform (WT). From the spectral aspect, noise reduction and baseline correction can be performed through algorithms such as MSC, SNV, Savitzky-Golay smoothing, first and second derivatives, FT, WT as well as orthogonal signal correction (OSC) [35,36].

The next step in the analysis flow will be image segmentation. Image segmentation serves to divide the pre-processed spectral image into different regions for the identification of region of interests (ROIs) [44]. In this process, segmentation algorithms are greatly preferred over manual segmentation due to the ease of operation and time saving. The selections of segmentation algorithms include thresholding (global thresholding or adaptive thresholding), morphological processing (erosion, dilation, open, close or watershed algorithm), edge-based segmentation (gradient-based or Laplacian-based methods) and spectral image segmentation [36].

Spatial analysis utilizing spectral image data usually involves quantitative measurement. In this process, gray-level object measurement is performed to quantify the intensity distribution of ROI extracted from image segmentation. Gray-level object measurements can be categorized according to intensity-based or texture-based measurements [45]. Intensity-based measurements are usually first-order measures such as mean [46,47], standard deviation, skew, energy and entropy [36]. Meanwhile, texture-based measurements are second-order measures such as joint distribution functions [36], gray-level co-occurrence matrix (GLCM) [46,48] and 2-D Gabor filter [49].

For spectral analysis, the data set will undergo multivariate analysis to reduce the spectral dimension and select the optimum wavelengths. Similar to the spectroscopy technique, some examples of multivariate analysis algorithms include PLS, linear discrimination analysis (LDA) [35,36], correlation analysis (CA) [50], PCA, ICA [41,51,52], ANN [53], sequential forward selection (SFS) [54] and GA [55]. These results from multivariate analysis will be used to develop calibration models for calibration, validation and prediction purposes [36].

The robustness of the final calibration model is evaluated from its ability to perform calibration and prediction. The calibration performance of the model is determined from the standard error of calibration (SEC), root mean square error of calibration (RMSEC) and the coefficient of determination ($r_{C}^{2}$) in the calibration set. Validation performance is determined via the root mean square error of cross-validation (RMSECV) and the coefficient of determination ($r_{V}^{2}$) in the validation set. Meanwhile, the prediction performance of the model is identified from standard error of prediction (SEP), root mean square error of prediction (RMSEP), residual predictive deviation (RPD) and the coefficient of determination ($r_{P}^{2}$) in the prediction set. Ideally, an effective model should register low SEC, RMSEC, RMSECV, SEP and RMSEP, with a minimum difference between SEC and SEP. Not least, higher $r_{C}^{2}$, $r_{P}^{2}$, $r_{P}^{2}$ and RPD are preferable [36].

#### 2.3.6. Pros and Cons of Spectral Imaging

This technique is advantageous as it omits chemical processes and requires minimum sample preparation. Moreover, the composition of multiple components in a sample can be simultaneously obtained. Upon spectral image acquisition, spectral imaging too allows the flexible selection of region of interest (ROI) for analysis. Furthermore, owing to the rich spatial and spectral information, spectral imaging can easily detect and differentiate subjects even though similar colors, overlapping spectra and morphological characteristics are present [36].

However, the spectral imaging technique does pose several limitations. Hardware speed is a major concern, especially in the case of hyperspectral imaging, due to the massive amounts of data to be acquired and analyzed. Moreover, spectral imaging includes the acquisition of redundant data, resulting in complex data analysis. Spectral imaging systems too require constant calibration in order to maintain their efficiency. The detection limits of spectral imaging are poorer compared to chemical-based analytical methods. Similar to spectroscopy, spectral imaging suffers from multicollinearity and requires multivariate analysis to address the issue. In addition, spectral imaging is inapplicable when the ROI is smaller than the size of a pixel or does not exhibit the characteristic spectral absorption. Lastly, spectral imaging may be irrelevant in the analysis of liquids and homogeneous samples since these samples do not pose distinctive and useful spatial information [36].

### 2.4. Technique Comparison

Table 4 presents a simple comparison of the optics and photonics techniques in agriculture that have been discussed earlier. From the comparison, the imaging technique is noted to be utilized for the extraction of spatial information only and is sensitive to small-sized objects. In contrast to the imaging technique, the spectroscopy technique allows acquisition of spectral information and is useful in accessing multi-constituent information. The spectral imaging technique covers the benefits of both imaging and spectroscopy techniques, allowing it to obtain spectral and spatial information simultaneously. Apart from this, spectral imaging has the added value of flexible spectral extraction as well as the capability of generating quality-attribute distribution. However, it should be noted that multispectral imaging has poorer access to spectral information compared to hyperspectral imaging due to the acquisition at limited number of wavelengths. Among the compared techniques, the spectral imaging technique can be said to be the most robust. Nonetheless, the area of application should be given the utmost consideration when selecting the best optics and photonic technique in order to avoid the underutilization or overutilization of a particular technique [36].

## 3. Optics and Photonics Applications in Agriculture

The optics and photonics techniques discussed above have been applied in various studies involving agricultural products. The studies will be tabulated in the following sections, enlisting details such as agriculture class, agriculture product, application area, wavelength details and country of applications.

### 3.1. Applications of Imaging Technique

Table 5 lists some of the agricultural works based on the imaging technique. The imaging technique is performed in the UV-VIS-IR range and involves the acquisition of spatial, color and thermal data from the inspected samples. These works show that the imaging technique is suited for inspection or analysis based on external features of the subject of interest. For instance, bruise detection [56,57] and disease detection [58,59] are performed by inspecting the external damage on the sample. In addition, quantitative analysis [60,61] is performed using the spatial information obtained. The color features extracted are also used for maturity evaluation [57,62,63] and nutrient content detection [64,65]. The thermal data, meanwhile, proves to be useful in similar occasions of bruise detection [66,67], disease detection [68,69] and maturity evaluation [70,71] by analyzing the temperature variations over the inspected sample. Not least, the most significant application of the imaging technique is the development of automated agricultural robots [72,73,74,75] and animal behavioral studies [76,77].

Based on Table 5, bruise detection, yield estimation, and disease identification are the three most common applications with imaging technique in agriculture. In bruise detection, a hyperspectral camera with broad operating wavelength from 400 to 5000 nm [78], a non-destructive and non-contact infrared sensing thermogram [66], and an infrared thermal imaging camera with high temperature resolution of 0.1 K [70] are among the instruments employed as the imaging technique. Moreover, the thermal camera with temperature resolution better than 0.5 °C [79], colour stereo vision camera which creates a 3D environment for further processing [86], and grading machine with a high accuracy of 96.47% [57] are employed for yield estimation. However, the grading machine proposed in [57] has a small capacity in estimation for 300 tomatoes per hour and does not efficiently work for tomato images with high specular reflection. In addition, infrared thermography is a popular device for disease identification due to its non-invasive monitoring and indirect visualization of downy mildew development [69]. This device takes the colour reflectance image for the detection of *V. inaequalis* development on apple leaves [68] and detects the pathogen in grapevines [84]. Additionally, an X-ray computed tomography scanner is utilized to obtain the cross-section of onion inoculated by pathogens [90], whilst an unmanned aerial vehicle (UAV) is presented to track the foliar disease in soybean [98].

Apart from the instrument, numerous types of algorithms are depicted in imaging technique. In bruise detection, PCA and a minimum noise fraction are proposed for 20 apple samples with threshold percentages of success within 86% to 93% and 87% to 97%, respectively [78]. In yield estimation, the fruit detection algorithm is presented for 8–120 apple samples with a correlation coefficient ranging from 0.83 to 0.88 [79]. In addition, the blob detector neural network is demonstrated to detect the yield estimation for both oranges and apples with intersection over union of 81.3% for orange and 83.8% for apple [61]. As for disease identification, a simple linear iterative clustering algorithm is presented for 3624 foliar images with high classification rate of 98.34% for height between 1 to 2 m [98]. The classification rate is reduced for approximately 2% for each meter from the examined height within 1 to 16 m. Moreover, an improved GoogLeNet and Cifar10 models are established for 500 images of maize leaf disease with 4:1 ratio for training and validation which allows the system to have a diversity of sample conditions [100]. The average identification accuracy of GoogLeNet and Cifar10 models is recorded as high as 98.9% and 98.8%, respectively. Apart from bruise detection, yield estimation, and disease identification, algorithms are also shown in maturity evaluation and acquisition of crop segmentation. In maturity evaluation, both a Fuzzy model [62] and medium filter algorithm [85] are employed for 3108 images on banana samples and 100 images on tomato samples with an average identification rate of 93.11%, and within 89% to 98%, respectively. The Fuzzy model is useful in handling ambiguous information for the banana fruit maturity detection using red-green-blue (RGB) components. In the acquisition of crop segmentation, K-mean clustering algorithm is presented for a clustering of apple samples with target acquisition rate of 84% [80]. This algorithm is commonly used in image segmentation whereby crop segmentation can be precisely attained, even with the presence of stems and leaves in the captured images.

### 3.2. Applications of Spectroscopy Technique

The applications of spectroscopy technique in agriculture are presented in Table 6. The spectroscopy technique is widely applied to inspect internal qualities that are externally invisible. A sizeable amount of research has performed spectroscopy in the UV-VIS-IR region to identify the internal constituents of agricultural products such as pigment compound in apple [111], moisture content in mushroom [112], protein and sugar in potato [113], and caffeine in coffee [114] among others. Within 400 to 1000 nm, 678 nm is sensitive to low chlorophyll content thus the reflectance at 678 ± 30 nm is suggested for the monitoring of the early stage of ripening and the pigment content change with a maximum correlation of closely 0.6 [111]. On the other hand, 590 to 700 nm is recommended for the maturity detection in early stage for yellow colour apple fruits with maximum correlation from 0.7 to 0.9. In the verification of moisture content in mushroom, the spectral region from 600 to 2200 nm gives the lowest standard deviation of cross validation as 0.644% and maximum correlation factor of 0.951 among the investigated wavelengths from 402 to 2490 nm [112]. A high experimental repeatability is presented by a standard deviation of 0.677% and a maximum correlation factor of 0.947 for a separate set of mushrooms of a similar type and treatment. In the protein and sugar content identification in potato, a modified PLS regression model is applied to calculate the relationship between the spectrum and chemical properties of the calibrated samples [113]. Based on the measurement, the standard deviations for crude protein, glucose, fructose, sucrose and red sugar for the 120 potato samples are 0.2%, 0.073%, 0.068%, 0.068%, and 0.122%, respectively. Correspondingly, the squared correlation coefficient for the above five parameters are deduced as 0.96, 0.70, 0.89, 0.62, and 0.82, respectively. In a total of 665 tea leaf samples, NIRS and liquid chromatography is coupled to a diode arrayed detector to determine its content of caffeine [114]. Among 375 calibration sets and 250 validation sets for caffeine in the tea leaf samples, a standard deviation of 8.6 and 8.9, as well as high squared correlation coefficients of 0.97 are acquired for both calibration and validation sets though regression model.

The quality and freshness of fruits [115,116], vegetables [117], and meat [118,119] can be easily inspected using spectroscopy as well. For instance, two wavelengths within 600 to 904 nm of VIS-NIR spectrum are investigated by correlation analysis to discriminate brown core and sound pears [115]. Using eight brown core pears and 32 sound pears, the percentage of soluble solid content achieves a precision of 97.8% and 99% within a standard deviation of 0.5% and 1%, respectively. In addition, NIR spectrum and PLS regression model are used to detect the total anthocyanins content (TAC) and total phenolic compounds (TPC) in jambu fruits [116]. With a total of 50 jambu samples scanning from 1000 to 2400 nm, the correlation coefficients of TAC and TPC are deduced as 0.98 and 0.94, and strong ratio to performance of deviation as 5.19 and 3.27, respectively. Besides that, a 250 to 350 GHz radiation is found to be suitable to distinguish the defective and proper sugar beet seeds [117]. A python package scikit-learn algorithm is used to determine the threshold for these two types of seeds, with 80% detection for proper seed and 94% detection for defective seed. Therefore, the average detection rate of this algorithm is 87%. In addition, meat fraud is injected into bovine meat, aiming to increase the water holding capacity. This issue is characterized with attenuated total reflectance Fourier transform infrared spectrum and the supervision of the 55 meat fraud adulterated samples through PLS square discriminant analysis [118]. The analysis records a precise detection as high as 91% of the adulterated samples. Apart from meat fraud, the freshness of mackerel fish is characterized with auto-fluorescence spectroscopy and analyzed with fluorescence excitation emission matrices (EEM) [119]. The fluorescence EEM data and real freshness values are modelled with PLS regression and an algorithm is developed for this smart system as a predictor with squared correlation coefficient of 0.89.

Furthermore, chemical residues in harvested product [120] or even plantation soil [121,122,123] can be easily identified, leading to easy detection on contamination of agricultural product. Residual pesticides such as phosmet and thiabendazole in apples are analyzed with surface-enhanced Raman spectroscopy (SERS) coupled with gold nanoparticle [120]. The sensitivities for detectable concentration are 0.5 µg/g for phosmet and 0.1 µg/g for thiabendazole. The PLS regression is also used to correlate the SERS spectrum with the concentration of pesticide in apples with squared correlation coefficient of 0936 for phosmet and 0.959 for thiabendazole. In addition, the effect of drying temperature on the nitrogen detection in soils at four different temperatures of 25 °C for placement, 50, 80, and 95 °C for drying is modelled based on NIR sensor and three successive algorithms, which are multiple linear regression, PLS, and competitive adaptive reweighed square on the spectral information [122]. Based on the three soil samples, loess, calcium soil, and black soil show the correlation coefficients of 0.9721, 0.9588, and 0.9486, respectively at the optimum drying temperature of 80°C. The detection of nitrogen in three types of soils is also alternatively performed in [123], with squared correlation coefficients of 0.95, 0.96 and 0.79 for loess, calcium soil and black soil using PLS regression model. The relatively lower squared correlation coefficient in black soil is due to the interference of high humus content and strong absorption of organic matter in black soil. Lastly, a point worth noting is that the dielectric and NMR spectroscopy are often adopted when the analysis involves more complex chemical compounds [124,125]. These complex chemical compounds include the vulcanization of natural rubber with sulfur-cured and peroxide-cured systems with different dynamics [124] and detection to changes in concentrations of pollutants in agriculture drainage such as heavy metal and heavy oxides [125]. In [124] a sulphur-cured system has features restricted segmented dynamics whereas peroxide-cured system has faster dynamics. In addition, network structure resulting from the vulcanization of both systems also influences the segmental dynamics of natural rubber. The peroxide-cured network is more homogeneous with spatial distribution of cross links than the sulphur-cured network with large inhomogeneity due to the presence of zinc oxide particles and the ionic interaction with the natural rubber chains. In [125], an X-ray fluorescence spectroscopy is employed to investigate the changes of pollutants in dried root and shoot plant parts at a temperature range from 30 to 90 °C. From the measurement, the concentration of pollutant is found to be higher in plant root than plant shoot through the analysis of frequency relaxation process via dielectric modulus measurement. Significantly, the removal of pollutants by plants will be enhanced upon subjecting them to a microwave heating power of 400 W for 30 min. Apart from the aforementioned applications, more applications of the spectroscopy technique in agricultural products with different methods and wavelengths are tabulated in Table 6.

### 3.3. Applications of Spectral Imaging Technique

As discussed earlier, the spectral imaging technique is a combination of both imaging and spectroscopy techniques. In the agriculture industry, both variations of hyperspectral and multispectral imaging are equally crucial, and some sample applications are compiled in Table 7. As observed from Table 7, hyperspectral imaging involves acquisition over a range of wavelengths while multispectral imaging involved acquisition at fewer selected wavelengths. The robustness of spectral imaging allows its usage in bruise detection [78,162,163], maturity evaluation [164,165,166,167,168], quality evaluation [169,170,171] and disease detection [172,173,174,175,176]. Internal attributes of samples [177,178,179] are easily acquired for analysis purposes as well.

In bruise detection of agriculture product, a machine vision system is integrated with optical filter at 740 and 950 nm to detect the bruise in rotating apple with a detection accuracy of 90 to 92% from 54 Pink Lady apples and 60 Ginger Gold apples [162]. In addition, the bruise detection in mushroom is carried out through a line scanning hyperspectral imaging instrument from 400 to 1000 nm with a spectroscopic resolution of 5 nm [163]. The PCA is applied to a set of data comprising of 50 normal and 50 bruise spectra, with standard deviation of 0.025 and 0.055, respectively. 

For the maturity evaluation of agriculture products such as peach fruit, a CCD camera is employed at 450, 675 and 800 nm, whereas the fruit ripening is characterized with the increasing in intensity from a histogram with a ratio of red divide with infrared red (R/IR) [164]. The firmness of the peach fruit reduces when the reflectance at 675 nm is increased. An analysis of variance (ANOVA) is presented to access the R/IR clustering which has the highest reflectance at 675 nm, and higher Fisher value as a function of higher R/IR ratio. Apart from the detection of maturity for peach fruit, the maturity stage of strawberry is detected using a hyperspectral imaging system from 380 to 1030 nm and from 874 to 1734 nm [165]. According to the PCA, the optimal wavelengths are from 441.1 to 1013.97 nm and from 941.46 to 1578.13 nm with a classification accuracy of above 85%. Moreover, the maturity of tomato is detected by an electromagnetic spectrum with a continuous 257 bands from 396 to 736 nm [166] and discrete band of 530, 595, 630, and 850 nm using a tomato maturity predictive sensor [167]. Based on the LDA, the classification error is reduced from 51% to 19% [166] and achieves detection accuracy above 85% [167]. The ripening in banana fruit is also characterized with a compact imaging system and an UAV from 500 to 700 nm [168]. The detection is based on the reflectance spectrum whereby in ripe banana, the main element is carotenoid which absorbs less light at 650 nm band. On the other hand, a green banana with a greater amount of chlorophyll than ripe banana absorbs more light at 650 nm band.

For the quality evaluation of agriculture product, the firmness test for two types of apple fruits is conducted with a laser-based multispectral imaging prototype which captures and processes four spectral scattering images at a speed of two fruits per second [169]. The multilinear regression models are developed using to predict the firmness of those two apple types at 680, 880, 905, and 940 nm with a correlation coefficient of 0.86. The quality of grape berries is determined by the reflectance spectrum from a hyperspectral imaging system, whilst the high reflectance at 500 nm, 660 to 700 nm, and 840 nm denotes the chlorophyll content, red-coloured anthocyanin pigment, and sugar content, respectively [170]. A partial least square regression (PLSR) model is applied in order to determine the correlation between the spectral information and the physico-chemical indices. The titratable acidity of the green and black grapes shows a coefficient of determination of 0.95 and 0.82, as well as soluble solid content of 0.94 and 0.93 at pH value of 0.8 and 0.9, respectively. The root mean square error for this method is 0.06 for green grape and 0.25 for black grape. Apart from fruits, the quality of tea leaves is classified by a hyperspectral imaging sensor at 762, 793, and 838 nm, supported by SVM algorithm [171]. Within 700 samples comprising of 500 training samples and 200 prediction sets, the SVM algorithm generates a total classification rate of 98% the training sample and 95% for the prediction set, at result of optimal regularization parameter of 4.37349 and kernel parameter of 13.2131 in SVM model.

For the disease detection in agriculture product, a fruit sorting machine is used to detect the citrus canker at 730 and 830 nm with a bandpass filter installed in the scanning camera [172]. A real-time image processing and classification algorithm is developed based on a two-band ratio (R830/R730) approach, which achieves a detection accuracy of 95.3% for 360 citrus samples. Next, a shortwave infrared hyperspectral imaging system is used to detect the sour skin in onion based on the suitable reflectance spectrum from 1070 to 1400 nm [173]. Two image analysis approaches are utilized based on the log-ratio images at two optimal wavelengths of 1070 and 1400 nm. A global threshold of 0.45 is integrated to segregate sour onion skin infection areas from log-ratio images. With Fisher’s discriminant analysis, the detection accuracy of 80% is achieved. The second image analysis approach is the incorporation of three parameters; max, contrast and homogeneity of the log-ratio images as the input features for the SVM. Subsequently, the Gaussian kernel generates higher detection accuracy as 87.14%. Apart from that, the tumorous chicken is detected by the combination of a CCD camera and imaging spectrograph from 420 to 850 nm [174]. Within the wavelength bands, the PCA select the three useful wavelengths of 465, 575, and 705 nm from the tumorous chicken image. Based on the images from 60 tumorous and 20 normal chicken, multispectral image analysis generates the ratio images, which are divided into ROI classified either as tumorous or normal chicken. The image features from ROI such as coefficient of variation, skewness and kurtosis are extracted as the input for the Fuzzy classifier, which generates the detection accuracy of 91% for normal chicken and 86% for tumorous chicken. To detect the nematodes in coffee cultivation, hyperspectral data is used in band simulation of the RapidEye sensor to determine the most sensitive spectral ranges for pathogen discrimination in coffee plants [176]. Multispectral classification identifies the spatial distribution of healthy, moderately infected, and severely infected coffee plants with an overall accuracy of 71%. Apart from the four main applications with the spectral imaging technique in agriculture products, more applications with different scanning methods and various wavelengths are tabulated in Table 7. 

## 4. Photonics Techniques Implementation in Food Safety Inspection and Quality Control

Food safety inspection and quality control is important for ensuring the high quality of agriculture products. To meet this criterion, photonics techniques have been extensively implemented into numerous applications. For instance, clean drinking water is undeniably one of the most important elements to sustain the organisms’ life. Contamination may happen when treated drinking water is travelling in the distribution system to the consumer, whilst the sensitivity to the inhibitor of contamination can be measured by the elevated dissolved organic matter (DOM) at the tap relative to the water leaving the treatment plant [216]. Across a biologically stable drinking water system, humic-like fluorescence (HLF) intensities of less than 2.2% relative standard deviation are measured after accounting for quenching by copper. In addition, a minor infiltration of a contaminant is detectable by sewage with a strong tryptophan-like fluorescence (TLF) signal thus validating the potential of DOM fluorescence in detecting the water quality changes in drinking water system. Moreover, fluorescence spectroscopy was demonstrated in evaluating the microbial quality of untreated drinking water through online monitoring [217]. The DOM peaks are targeted at excitation and emission wavelengths of 280 and 365 nm for TLF, as well as 280 and 450 nm for HLF. Both TLF and HLF are strongly correlated to micro-bacterial cells such as *E. coli* with a correlation coefficient of 0.71 to 0.77. In comparison to turbidity for *E. coli* with correlation coefficient of only 0.4 to 0.48, the DOM sensor appears to be a better indicator for micro-bacterial cells in untreated drinking water. Apart from the DOM sensor, an optical sensor was proposed to differentiate the particles in drinking water as either bacteria or abiotic particles with an accuracy of 90 ± 7% and 78 ± 14% for monotype and fix-type suspensions, respectively, based on a 3D image recognition and classification algorithm [218]. In addition, this optical sensor can detect micro-particles with minimum size of 0.77 μm. Significantly, the aforementioned optical sensors incorporating photonic techniques serve as an early warning for drinking water pollution. 

Photonics and optics have also recently gained popularity in the quality inspection of food product. This is because food inspection in the production line needs to be carried out at fast speed and a very fast monitoring system is needed. Food inspection can become even more challenging when it is dealing with large quantities of sample moving very quickly on the conveyor belt. Therefore, high speed and high sensitivity optical system will be very suitable for the online monitoring and inspection of food product. For example, research work on UV-visible-NIR optical spectroscopy have been carried out extensively in the monitoring of extra virgin olive oil [219], honey [220], tea [221], dairy product [222] and alcoholic beverages [223]. However, as these works focus on wavebands below 1100 nm, the results and consistency of the conclusions may be easily affected by ambient lighting conditions and the change in color of the beverages or product. Therefore, more research work shall be carried out to characterize these food products in the NIR (>1100 nm) and MIR wavebands in order to obtain the optical “fingerprint” that correlates to the quality and food safety level of the product.

In addition, food preservative exceeding the allowable limit has been a critical issue in ensuring the health of the public. Butylated Hydroxytoluene (BHT) is commonly used as an antioxidant agent in canned food or bottled beverages. Several optical sensing techniques such as optical spectroscopy and fluorescence may be able to detect the concentration of BHT. BHT is also commonly known as 2,6-ditertiarybutyl-para-cresol (DBPC). Recently, Leong et al. [224] reported the detection of DBPC in transformer oil using optical spectroscopy at waveband near to 1403 nm. This opens up the opportunity of detecting the concentration of BHT in canned food or bottled beverages, leading to an online monitoring system that uses the optical spectroscopy method. 

Due to the lack of attention paid during the preparation processes or due to the contamination of water and environment, hazardous residual materials are occasionally found in food. These hazardous materials include heavy metal, pesticides and antibiotics. Conventionally, the screening process or food safety inspections were carried out using laboratory-based equipment or measurement methods such as gas chromatography (GC), GC-mass spectrometry and high-performance liquid chromatography [225]. However, these methods only allow inspection based on sampling due to the high cost and long result waiting time. In this context, optical detection methods such as optical spectroscopy, Raman spectroscopy and fluorescence can be explored for their possible utilization in the online monitoring of food products in order to ensure that they are free of hazardous residual materials. 

## 5. Photonics Techniques Implementation in Tropical Countries Agriculture

Blessed with wide spans of fertile soil, rich marine ecology, abundant rainfall and a tropical climate, tropical countries are exceptionally suited for a myriad of agricultural activities [226,227,228]. Agriculture activities boost the country’s economy by supplying food sources and industrial raw materials. This sector also provides income to farmers, raising their living standards in rural areas. An example of tropical countries with active agricultural activities is Malaysia. Dating back to the early years following Malaysia’s independence in 1957, the agriculture sector has been a signifficant driver towards socio-economic development in Malaysia. However, in the early 1980s, the growth of the agricultural field came to an abrupt halt due to the sharp decline in commodity prices, limited technical specialty, volatile rubber prices and lack of incentives [229,230]. Industrialization soon became the leading economic sector, with great focus directed towards manufacturing and services [230]. Fortunately, the agriculture sector is once again emphasized upon the Asian financial crisis in 1997, acting as a measure to minimize external economic shock by first strengthening the domestic economy [227,231]. Since then, agriculture has always been a major agenda item of Malaysian economic plans, with a recent target directed towards modernizing agriculture as drafted in the Eleventh Malaysia Plan [232]. To date, amidst industrial developments, Malaysia has approximately 4.06 million hectares of agricultural land, with 80% allocated for commercial crops such as palm oil, rubber, cocoa, coconut and pepper [229,233], while a portion of the remaining 20% was utilized for the cultivation of agro-food crops [226]. These remarkable statistics have validated the potential of a tropical country to develop its agricultural sector. Apart from Malaysia, other tropical countries such as Indonesia and Thailand are also actively involved in agriculture activities. The following sections will discuss some of the agricultural crops in tropical countries in which optics and photonic techniques can be easily integrated for automated plantation management, yield increment, quality inspection and disease control.

### 5.1. Implementation in Palm Oil-Related Activities

Palm oil is an extremely valuable commercial crop in tropical countries. Palm oil, which is extracted from oil palm, is often used as raw materials for the production of biofuel, biofertilizers, oleochemicals, biomass products, nutraceuticals and pharmaceuticals. In fact, tropical countries are among the global leaders in the palm oil industry [234]. The implementation of optics and photonic techniques in palm oil-related activities will maintain the competitive power of the tropical countries in the field and help to reap the associated economic benefits.

The implementation of optics and photonics techniques in oil palm related activities can start from the development of agriculture robots. The development of an agriculture robot involves the implementation of the imaging technique in its operation. Spatial and color information attained by the agriculture robot through the imaging technique will greatly improve the efficiency of palm oil plantation management. Automated palm oil fruit harvest is potentially applicable by pinpointing the fruit position as presented in [72,74,75] for other crops. Besides, automated weed detection and removal [73] as well as automated fertilizing can be performed using the developed agriculture robot.

In addition, palm oil quality is governed by fatty acid, moisture and peroxide contents. Microbial or oxidation reactions that take place during the storage of oil palm fruit may modify these contents, resulting in a depreciation of palm oil quality [235]. Under common operations, palm oil plantations are usually distanced further away from refinery factories. Bulk transport of palm fruit upon reaching the necessary processing quota is often practiced for cost savings. As a result, palm fruits that have been harvested earlier will be stored in dedicated storage spaces. The time difference between harvesting and processing greatly increases the risk for microbial or oxidation reaction to take place. In this scenario, spectroscopy or spectral imaging can be implemented in the palm oil extraction stage to perform oil quality segregation. This will greatly prevent contamination of low-quality palm oil in further downstream processes, promoting process efficiency and increasing palm oil yield.

Another area in which optics and photonics techniques may be applied for oil palm activities is disease detection. The most devastating diseases that attack palm oil plantations in South East Asia are basal stem rot (BSR) and upper stem rot (USR). These diseases result in certain death of oil palms if not controlled effectively, resulting in yield loss and disrupting the plantation cycle. These fatal diseases are identified to be caused by the *Ganoderma boninense* (*G. boninense*) fungus. However, the identification of the root cause of these diseases is still insufficient as they cannot be controlled even with the slightest delay in infection detection [236]. In this area, spectral imaging presents itself as one of the possible alternatives to perform early detection of the *G. boninense* fungus [237]. Samples of suspicious fungi in the palm oil plantation can be simultaneously collected and analyzed to identify the presence of disease-causing *G. boninense*. From here, preventive measures can be effectively performed to curb any possible disease spreading.

### 5.2. Implementation in Natural Rubber Related Activities

Natural rubber is an important commodity that finds it place in the manufacturing of various household, industrial and medical products. Rubber tree plantations have been widely established in the fertile soils of tropical countries. The usage of optics and photonic techniques will again prove to be beneficial in this area.

The simplest idea will again start from the usage of agriculture robots during the plantation stage. In the context of rubber tree plantations, the imaging technique will provide visual guidance for the agriculture robots to perform the scheduled collection of field latex. The usage of these robots will gradually replace manual latex collection done by rubber tappers. This approach will address the decline in manpower to maintain rubber tree plantations.

Meanwhile, the spectroscopy technique can be utilized in the later rubber processing stages. The first application would be rubber quality grading. For instance, cup lump raw rubber, which is an important material in tires, seal strips, conveyor belts and other moulded rubber products, can be graded by using VIS-NIR spectroscopy to inspect the moisture content of the rubber. This spectroscopic approach is fast, accurate and more reliable compared to manual inspection through sight and touch [155]. Similarly, the protein and lipid contents in natural rubber can be detected through NIR-MIR spectroscopy to enable grading [152]. Lastly, spectroscopy variations, such as NIR-MIR, Raman, dielectric or NMR, can be opted to study the structure and properties of rubber during vulcanization. Such studies allow the analysis and selection of accelerators, activators and retarders, leading to improved characteristics in the vulcanized rubber and an optimized vulcanizing process [124,153].

### 5.3. Implementation in Agro-Food Crops Related Activities

It is important to increase food production and achieve a self-sufficiency level (SSL) for a growing country to become an advanced country. Currently, the agro-food crops in tropical countries comprise of grains, organic fruits and vegetables, herbs and spices, livestock and fisheries [232,234]. By referring to some of the applications stated in Section 3.1, Section 3.2 and Section 3.3, optics and photonic techniques can once again improve the overall quality and yield of these crops.

Starting from grains such as rice and corn, crop harvest [72] and weed removal [73] can be easily performed by agriculture robots with imaging capabilities. Thermal imaging can be conducted to evaluate water stress in crops for irrigation control [97]. Moreover, the development of mobile phone application to perform color-based identification of nitrogen content in rice and corn plant is another interesting idea. The usage of such applications promotes the portable and on-site analysis of fertilizer requirements in crop fields [64].

At the same time, all three optics and photonic techniques discussed earlier can be fully utilized to inspect the harvested organic fruits and vegetables for quality evaluation. For instance, imaging in either VIS or IR region is useful in detecting external damage or bruises in mangosteens, wax jambus, cherry tomatoes and more. Spectroscopy may be performed as well to inspect internal features or maturity of fruits and vegetables. Not least, spectral imaging may be considered when spatial and spectral information are required simultaneously for quality evaluation. Meanwhile, the quality inspection of meat products, such as chicken, beef, lamb, and fish among others, is strongly preferred to be performed using spectroscopy or spectral imaging. These two techniques are suitable for identifying the microbial spoilage of meat products due to their ability to obtain spectral information. With the integration and application of optics and photonics in the agriculture industry, it is anticipated that the agricultural products in the tropical countries will meet the public expectation of higher food quality.

### 5.4. Possible Challenges

The prevailing research challenges of integrating optics and photonics techniques into the agriculture field are the reliability issue of the laser source and sensor, effect of the ambient environmental condition into optics system, and expensive semiconductor materials at operating wavelength from short to mid-IR range. First and foremost, the illumination intensity of the laser and the sensitivity of the sensor may change over time, which leads to the need for recalibration of the system. Therefore, more research is required in terms of the design and fabrication of a more reliable laser source, sensor and optical detector. In addition, the effect of the ambient condition such as humidity, surrounding temperature, and dust particles could be a hindrance in ensuring consistent results obtained from the optical system. Hence, research into the minimization of these effects on the optical system is significant to improve the system performance such as higher sensitivity, lower systematic error and maintenance rate. Moreover, silicon is well-known for its optimum wavelength operation below 1000 nm. From short to mid-IR range, examples of more viable semiconductor materials are gallium antimonide and indium gallium arsenide. The investigation in terms of generating a higher efficiency using these materials for a cost-effective solution creates the research opportunities for further exploration in both simulation and experimental works. 

Apart from the research challenges, the main challenge in introducing the discussed optics and photonic techniques into the field of agriculture in tropical countries would be gaining the acceptance of farmers, fisherman and smallholders. The introduction of modern technology and new agriculture practices often raises concerns surrounding their technical and economic feasibilities. Farm and plantation owners will prefer traditional agriculture practices as newly introduced technologies are often regarded to be more suited to a controlled laboratory environment. In this scenario, technology vendors should ensure that complete field testing has been done in the environment where the technology will be introduced. A probationary period may also be set to allow owners to try out and experience the benefits brought forth by the proposed technologies.

The next challenge would be on financial limitations. In general, the cost to fully implement optics and photonics techniques in existing agriculture activities may be a burden to the owners, especially those involving sophisticated optical tools. This deterring factor may be mitigated if financial aids are provided to the owners. In this case, the government of tropical countries should set the right path by providing funds to the owners through attractive policies. For instance, a loan policy of flexible repayment based on harvest cycles is more attractive compared to one of fixed term financing since owners are now presented with flexible loans [232].

Lastly, another challenge lies with the need of technical support. When introducing the optics and photonic techniques, technical training should be provided to farm and plantation workers in order to familiarize them with the operations of new tools. At the same time, advisory and technical services should be easily available in case the agriculture tools experience downtime or require scheduled maintenance.

## 6. Conclusions

In conclusion, optics and photonics exhibit great benefits if they are integrated into the agricultural industry. A complete knowledge of the behaviors and properties of light upon light-material interaction allows the quantitative and qualitative analysis of agriculture products. In general, optics and photonic techniques for agricultural purposes can be categorized into imaging, spectroscopy and spectral imaging techniques. The imaging technique is effective in collecting spatial, color and thermal information, whereas the spectroscopy technique is essential for collecting spectral information. Meanwhile, spectral imaging is a combination of both imaging and spectroscopy techniques, allowing the collection of a complete data set. These three optics and photonic techniques have been utilized in agriculture categories such as fruits, vegetables, grain, meat, dairy produce, oil, beverages, and commercial crops, as well as farm and plantation management. These works can be referred to and emulated in the agriculture industry of tropical countries, especially in agriculture activities related to oil palm, rubber and agro-food crops. However, challenges in terms of public acceptance, finance and technical support should be overcome before achieving a complete integration of optics and photonics techniques in the agriculture industry.

Thus, the key contribution of this study is the comprehensive analysis of different optics and photonics systems in agricultural applications to provide a detail idea of the advanced techniques and their future deployment in agriculture cultivation and harvesting. The review has proposed important and selective suggestions for the further technological development of optics and photonics in future agricultural applications:The incorporation of optical sensors into photonics detection techniques that serve as an early warning for drinking water pollution.The characterization of canned food or bottled beverages in the NIR (>1100 nm) and MIR wavebands for their optical “fingerprint” that correlates to the quality and food safety level of the product, such as preservatives concentration.The characterization on hazardous residual materials in food using optical spectroscopy, Raman spectroscopy and fluorescence.The implementation of an agricultural robot to perform better palm oil plantation management, scheduled collection of field latex and weed removal.The spectral imaging provides early detection of disease-causing *G. boninense* in the oil palm.Spectroscopy provides moisture content inspection, protein and lipid content detection, as well as improving the rubber vulcanizing process.The imaging technique detects external damage or bruises on organic fruits and vegetables.

## Figures and Tables

**Figure 1 molecules-24-02025-f001:**
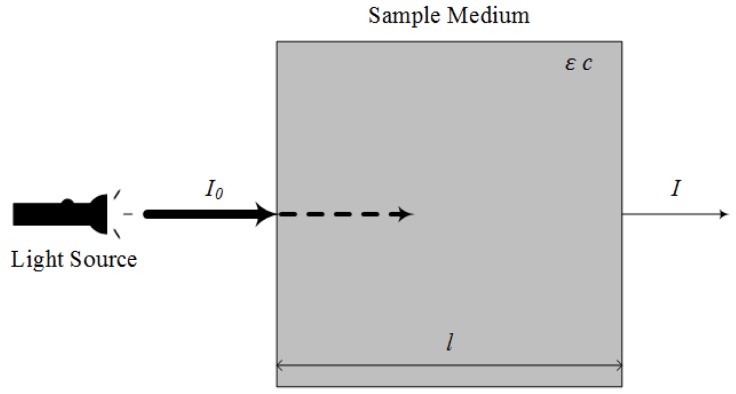
Model of Beer-Lambert’s Law [18].

**Figure 2 molecules-24-02025-f002:**
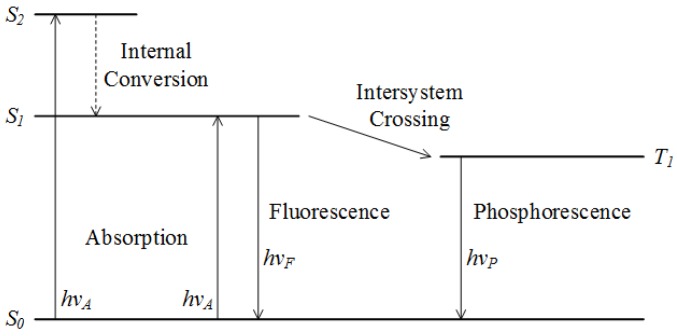
Jablonski diagram [17]. Reproduced with permission from A. Nawrocka, Advances in Agrophysical Research, Published by IntechOpen, 2013.

**Figure 3 molecules-24-02025-f003:**
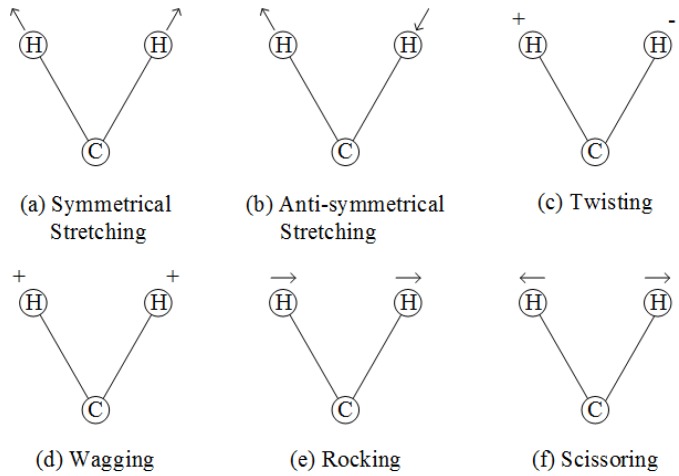
Vibrations in diatomic molecules [17]. Reproduced with permission from A. Nawrocka, Advances in Agrophysical Research, Published by IntechOpen, 2013.

**Figure 4 molecules-24-02025-f004:**
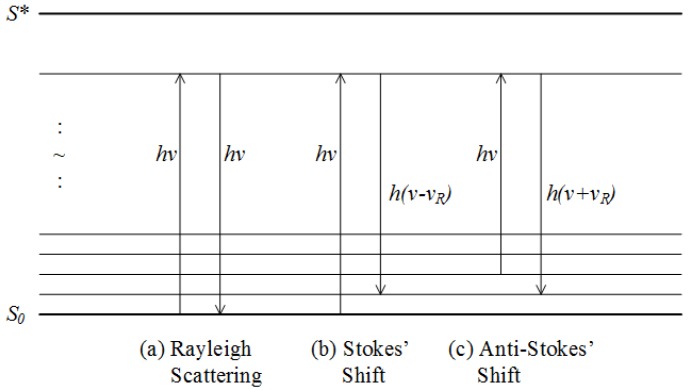
Light scattering schemes [17]. Reproduced with permission from A. Nawrocka, Advances in Agrophysical Research, Published by IntechOpen, 2013.

**Figure 5 molecules-24-02025-f005:**
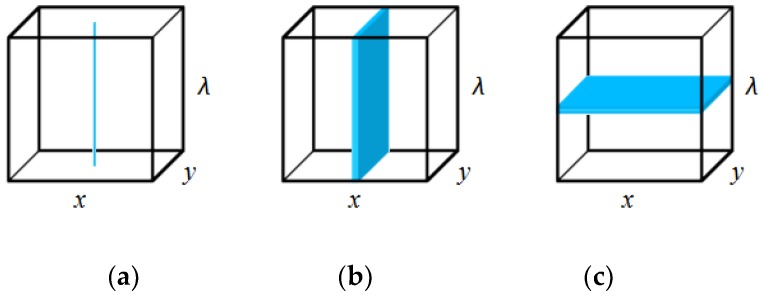
Methods of spectral image acquisitions with (**a**) point scan, (**b**) line scan, and (**c**) area scan [35]. Reproduced with permission from J. Qin, Journal of Food Engineering, Published by Elsevier, 2013.

**Figure 6 molecules-24-02025-f006:**
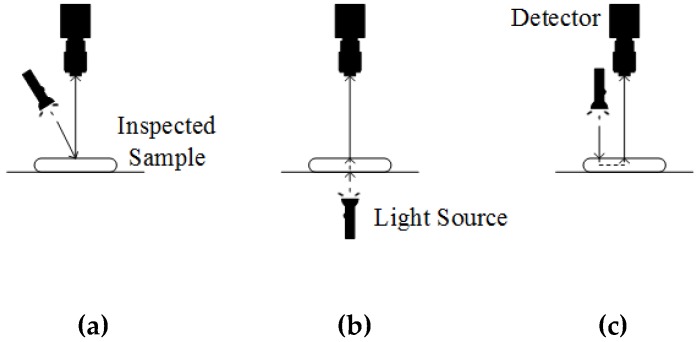
Sensing modes in spectral imaging; (**a**) reflectance, (**b**) transmittance, and (**c**) Interactance [36]. Reproduced with permission from D. Wu, Innovative Food Science & Emerging Technologies, Published by Elsevier, 2013.

**Table 1 molecules-24-02025-t001:** Examples of NIR absorption bands [29].

Wavelength (nm)	Wavenumber (cm^−1^)	Assignment
		Water
1454	6878	1st overtone O–H stretching
1932	5176	O–H combination
		Proteins
1208	8278	2nd overtone C–H stretching
1465	6826	1st overtone N–H and O–H stretching
1734	5767	1st overtone C–H stretching
1932 2058 2180	5176 4859 4587	N–H combination and O–H stretching
2302 2342	4344 4270	C–H stretching combination
		Oil
1210	8264	2nd overtone C–H stretching
1406	7112	1st overtone N–H and O–H stretching
1718 1760	5821 5682	1st overtone C–H stretching
2114	4730	N–H combination and O–H stretching
2308 2346	4333 4263	C–H stretching combination
		Starch
1204	8306	2nd overtone C–H stretching
1464	6831	1st overtone N–H and O–H stretching
1932 2100	5176 4762	N–H combination and O–H stretching
2290 2324	4367 4303	C–H stretching combination

**Table 2 molecules-24-02025-t002:** Examples of MIR absorption bands [27].

Wavelength (nm)	Wavenumber (cm^−1^)	Assignment
		**Water**
2.778–3.125	3200–3600	O–H stretching
6.061	1650	H–OH stretching
		**Proteins**
5.917–6.250	1600–1690	Amide I (C=O stretching)
6.349–6.757	1480–1575	Amide II (C–N stretching and N–H bending)
7.692–8.130	1230–1300	Amide III (C–N stretching and N–H bending)
		**Fats**
3.333–3.571	2800–3000	C–H stretching
5.731–5.797	1725–1745	C=O stretching
10.309	970	C=C–H bending
		**Carbohydrates**
3.333–3.571	2800–3000	C–H stretching
7.143–12.500	800–1400	Skeletal stretching and bending

**Table 3 molecules-24-02025-t003:** Examples of Raman bands [30].

Wavelength (nm)	Wavenumber (cm^−1^)	Assignment
		**Water**
2.778–3.125	3200–3600	O–H stretching
		**Proteins**
19.608 19.048 18.349	510 525 545	S–S stretching
14.925–15.873 13.423–14.286	630–670 700–745	C–S stretching
5.882–6.250	1600–1700	Amide I (C=O stretching and N–H bending)
8.032–8.097	1235–1245	Amide III (C–N stretching and N–H bending)
3.876–3.922	2550–2580	S–H stretching
3.333–3.571	2800–3000	C–H stretching
		**Fats**
6.940	1441	CH_2_ bending
6.863	1457	CH_3_–CH_2_ bending
6.039	1656	C=C stretching
3.378–3.503	2855–2960	C–H stretching
		Carbohydrates
11.962	836	C–C stretching
9.398	1064	C–O stretching
3.434 3.397	2912 2944	C–H stretching
2.898	3451	O–H stretching

**Table 4 molecules-24-02025-t004:** Comparison of optics and photonics techniques in agriculture [36].

Characteristics	Imaging	Spectroscopy	Spectral Imaging
Spectral information	×	✓	✓
Spatial information	✓	×	✓
Multi-constituent information	×	✓	✓
Sensitivity to small-sized objects	✓	×	✓
Flexibility of spectral extraction	×	×	✓
Generation of quality-attribute distribution	×	×	✓

**Table 5 molecules-24-02025-t005:** Applications of imaging technique in agriculture.

Class	Product	Application	Ref.
Fruit	Apple	Bruise detection (thermal)	[66,67,70,78]
	Apple	Maturity evaluation (thermal)	[70]
	Apple	Yield estimation (thermal)	[79]
	Apple	Scab disease detection (thermal)	[68]
	Green apple	Acquisition of segmented fruit region	[80]
	Green apple and orange	Yield estimation	[61]
	Orange	Texture analysis	[81]
	Orange	Bruise detection (thermal)	[67]
	Citrus	Water stress evaluation (thermal)	[82]
	Pear	Maturity evaluation (thermal)	[71]
	Banana	Maturity evaluation	[62]
	Banana	Maturity evaluation	[63]
	Persimmon	Maturity evaluation (thermal)	[71]
	Passion fruit	Mass and volume estimation	[83]
	Blueberry	Bruise detection	[56]
	Grapevine	Pathogen detection (thermal)	[84]
	Tomato	Fruit detection	[85,86]
	Tomato	Bruise detection and maturity evaluation	[57]
	Tomato	Bruise detection (thermal)	[87]
	Tomato	Maturity evaluation (thermal)	[71]
	Tomato	Clustered fruit detection	[88]
	Sweet peppers	Peduncle detection	[89]
	Onion	Post-harvest quality assessment (thermal)	[90]
	Lettuce	Segmentation of vegetable	[91]
	Cucumber	Downy mildew disease detection (thermal)	[69,92,93]
Grain	Rice leaf	Nitrogen content detection	[64]
	Wheat	Yield estimation (thermal)	[94,95,96]
	Corn	Water stress evaluation (thermal)	[97]
	Macadamia nuts	Yield estimation	[60]
	Soybean	Identification of foliar disease	[98]
	Soybean	Identification of leaf disease	[59]
	Maize	Yield estimation (thermal)	[99]
	Maize	Identification of leaf disease	[100]
	Maize	Cultivar identification	[101]
Commercial	Cotton	Water stress evaluation (thermal)	[97,102]
	Silkworm	Gender identification	[103]
Farm and Plantation	Seed	Viability evaluation (thermal)	[104]
	Wheat field	Estimation of nutrient content	[65]
	Cauliflower plantation	Weed detection	[73]
	Asparagus plantation	Crop harvest robot vision	[74]
	Sugar beet and rape plantation	Agriculture robot vision	[75]
	Grapevines	Estimation of intra-parcel grape quantities	[105]
	Cow farm	Behavioural studies	[76,106]
	Goat and sheep farm	Animal species identification	[107]
	Fish aquarium	Behavioural studies	[77,108]
	Baby shrimp farm	Chlorine level detection	[109]
	Orchid farm	Disease and pest detection	[58]
	Surface and ground water	Chemical content detection	[110]

**Table 6 molecules-24-02025-t006:** Applications of spectroscopy technique in agriculture.

Class	Product	Application	Method	Wavelength (nm)	Ref.
Fruit	Apple	Pigment content change during ripening	UV-VIS-NIR	400–1000	[111]
	Apple	Soluble solid content detection	VIS-NIR	500–1100, 1000–2500	[33]
	Apple	Pesticide residue detection	Raman	5–18 µm	[120]
	Pear	Brown core and soluble solid content detection	UV-VIS-NIR	200–1100	[115]
	Mango	Maturity evaluation	NIR	1200–2200	[126]
	Peach	Peach variety identification	NIR	833–2500	[127]
	Wax jambu	Quality inspection	NIR	1000–2400	[116]
	Grape leaf	Water content estimation	UV-VIS-NIR	350–2500	[128]
Vegetable	Carrot	Carotenoid, fructose, glucose, sucrose and sugar content detection	NIR	1108–2490	[129]
	Potato	Bruise detection	UV-VIS-NIR	250–1750	[130]
	Potato	Protein, fructose, glucose, starch and sucrose content detection	NIR	1100–2500	[113]
	Onion	Soluble solid content detection	VIS-NIR	500–1200	[131]
	Oilseed rape leaf	Aspartic acid content detection	NIR	1100–2500	[132]
	Sugar beet seeds	Quality control	Time-domain spectroscopy	250–350 GHz	[117]
	Mushroom	Moisture content detection	VIS-NIR	600–2200	[112]
Grain	Corn seed	Viability evaluation	NIR Raman	1000–2500 3.125–59 µm	[133]
	Almond	Internal defect detection	VIS-NIR	700–1400	[134,135]
	Maize	Identification of transgenic ingredients	THz spectral	0–4.5 THz	[136]
	Rice, maize and peanut	Germination and growth of crop	UV-VIS FTIR	380.85–796.62 nm 562.72–3865.11 cm^−1^	[137]
Meat	Beef	Thermal change inspection	Fluorescence	250–550	[138]
	Beef	Adulteration detection	NIR-MIR	2.5–19 µm	[118]
	Frozen fish	Freshness evaluation	Fluorescence	250–800	[119]
Dairy	Egg	Contamination detection	UV-VIS-NIR	200–860	[139]
	Goat milk	Fatty acid content detection	VIS-NIR	400–2498	[140]
Oil	Edible oil	Stability analysis	NMR	300 MHz (1H)	[141]
	Olive oil	Adulteration detection	Fluorescence	250–720	[142]
	Ocimum essential oil	Antioxidant property identification	NIR-MIR	2.5–18 µm	[143]
Beverage	Tea leaf	Tea polyphenol level detection	UV-VIS-NIR	347–2506	[144]
	Green tea leaf	Caffeine and catechins content detection	VIS-NIR	400–2500	[114]
	Coffee	Geographic and genotypic origin identification	NIR	1100–2498	[145]
	Coffee	Roasting degree and blend composition detection	NIR	800–2857	[146]
	Tomato juice	Quality inspection	NIR-MIR	2.5–14 µm	[147]
	Apple wine	Volatile compound detection	NIR	833–2500	[148]
	Rice wine	Fermentation monitoring	NIR-MIR	2.5–25 µm	[149]
Commercial	Cotton fibre	Cotton type identification	NIR	800–2500	[150]
	Cotton fibre	Cotton fibre micronaire measurement	VIS-NIR	400–2500	[151]
	Natural rubber	Protein and lipid content detection	NIR-MIR	2.5–25 µm	[152]
	Natural rubber	Chemical interaction during vulcanizing process	NIR-MIR Raman	2.5–25 µm 3.125–100 µm, 6.25–50 µm	[153]
	Natural rubber	Rubber silane reaction	NMR	400 MHz (1H), 100.6 MHz (13C)	[154]
	Natural rubber	Moisture content detection	VIS-NIR	400–1100	[155]
	Natural rubber	Vulcanization system effect	Dielectric NMR	10-1 < Hz < 107 20 MHz (1H)	[124]
	Neem leaf	Pest control	UV-VIS FTIR XRD	200–800 nm 250–4000 cm^−1^ 10–80°	[156]
Farm and Plantation	Soil	Quality inspection	NIR	780–5000	[157]
	Soil	Nitrogen content detection	NIR	800–2564	[158]
	Soil	Chemical and physical property estimation	NIR-MIR	1430–2500, 2.5–27 µm	[159]
	Soil	Nitrogen detection	NIR	900–1700	[122]
	Soil	Nitrogen detection	NIR	900–1700	[123]
	Soil and water	Contaminant detection	VIS-NIR	400–2500	[121]
	Water hyacinth Soybean straw	Pollutant concentration detection Detection of biomass	Dielectric Fluorescence Near infrared spectroscopy	10-1 < Hz < 106 N/A 4000–12,000 cm^−1^	[125] [160]
	Flower	Plant type identification	VIS	635, 685, 785	[161]

**Table 7 molecules-24-02025-t007:** Applications of spectral imaging technique in agriculture.

Class	Product	Application	Method	Wavelength (nm)	Ref.
Fruit	Apple	Bruise detection	Hyper. line scan	400–2500, 1000–2500	[78,180]
	Apple	Bruise detection timing	Hyper. line scan	400–2500	[181]
	Apple	Bruise detection	Multi. area scan	740, 950	[162]
	Apple	Bruise and faeces detection	Multi. line scan	530, 665, 750, 800	[182]
	Apple	Firmness evaluation	Multi. area scan	680, 880, 905, 940	[169]
	Citrus	Canker detection	Multi. area scan	730, 830	[172]
	Peach	Firmness evaluation	Hyper. line scan	500–1000	[183]
	Peach	Maturity evaluation	Multi. area scan	450, 675, 800	[164]
	Cantaloupe	Faeces detection	Hyper. line scan	425–774	[184]
	Blueberry	Firmness evaluation, soluble solid content detection	Hyper. line scan	400–1000	[177,185]
	Strawberry	Maturity evaluation	Hyper. line scan	380–1030 874–1734	[165]
	Cherry	Pit detection	Hyper. line scan	450–1000	[186]
	Grape	Quality evaluation	Hyper. line scan	400–1000	[170]
	Banana	Maturity evaluation	Hyper. area scan	500–700	[168]
	Tomato	Maturity evaluation	Hyper. line scan	396–736	[166]
	Tomato	Maturity evaluation	Multi. area scan	530, 595, 630, 850	[167]
	Cucumber	Chilling injury detection	Hyper. line scan	447–951	[187]
Vegetable	Freeze-dried broccoli	Glucosinolate detection	Hyper. line scan	400–1700	[188]
	Potato	Cooking time prediction	Hyper. line scan	400–1000	[189]
	Onion	Sour skin disease detection	Hyper. area scan	950–1650	[173]
	Mushroom	Bruise detection	Hyper. line scan	400–1000	[163]
Grain	Rice plant	Nitrogen content detection	Hyper. line scan	400–1000	[190,191]
	Thai jasmine rice	Rice variety identification	Multi. area scan	545, 575	[192]
	Wheat	Fungus detection	Hyper. area scan	1000-1600	[193]
	Wheat	Damage detection	Hyper. line scan	1000–2500	[194]
	Peanut	Tomato spot wilt disease detection	Multi. Area scan	475, 560, 668, 717, 840	[195]
	Corn	Oil and oleic acid content detection	Hyper. area scan	950-1700	[196]
	Corn	Aflatoxin detection	Hyper. line scan	400–600	[197]
Meat	Chicken	Skin tumour detection	Hyper. line scan	420–850	[174]
	Chicken	Heart disease detection	Multi. area scan	495, 535, 585, 605	[198]
	Chicken	Faeces detection	Multi. area scan	520, 560	[199]
	Chicken	Wholesomeness inspection	Multi. line scan	580, 620	[200]
	Beef	Tenderness evaluation	Hyper. line scan	400–1000	[201]
	Beef	Microbial spoilage detection	Hyper. line scan	400–1100	[202]
	Lamb	Lamb variety identification	Hyper. line scan	900–1700	[203]
	Pork meat	*E. coli* detection	Hyper. line scan	470–960	[204]
	Pork meat	Quality inspection	Hyper. line scan	900–1700	[205]
	Fish	Moisture and fat content detection	Hyper. line scan	460–1040	[206]
	Fish	Ridge detection	Hyper. line scan	400–1000	[207]
	Salmon	Microbial spoilage detection	Hyper. line scan	400–1000 880–1720	[208]
	Dehydrated prawn	Moisture content detection	Hyper. line scan	380–1100	[209]
	Prawn	Adulteration detection	Hyper. line scan	380–1030 900–1700	[210]
Dairy	Milk powder	Melamine detection	Hyper. line scan	990–1700	[211]
	Milk	Fat content detection	Hyper. line scan	530–900	[178]
	Milk	Melamine detection	Hyper. point scan	4–98 µm	[212]
Oil	Olive oil	Free acidity, peroxide and moisture content detection	Hyper. line scan	900–1700	[179]
Beverage	Tea	Quality inspection	Hyper. line scan	408–1117	[171]
	Tea	Moisture content detection	Hyper. line scan	874–1734	[213]
	Tea	Tea variety identification	Multi. area scan	580, 680, 800	[214]
Farm and Plantation	Tea bush	Tea variety, growth status and disease identification	Hyper. area scan	325–1075	[175]
	Coffee crop	Detection of disease/infection	Hyper. area scan	440–850	[176]
	Coffee plantation	Monitoring chlorophyll content	Multi. area scan	490–2190	[215]

Note: Hyper. = hyperspectral, multi. = multispectral.

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
