# Peer review of "Applications of Photonics in Agriculture Sector: A Review"

_molecules, 2019, doi:10.3390/molecules24102025_

Round 1
Reviewer 1 Report
Could you please discuss the possibility of useing computer image analysis technik for analysis of different agriculture processes as: mixing of agriculture products, drying, evaluation of quality work of field machinery during crop planting etc.
Please correct english - some letters are missing.
Author Response
We would like to sincerely thank you very much for taking time, out of your busy schedule to manage the review process. We would equally wish to thank reviewers for their considerations. We have carefully considered all the comments made by reviewers appertaining to the above article and made the following changes to clarify some of his/her concerns and thus improve the overall presentation of the paper, and its quality and clarity.
The details of modifications and our replies are as follows:
1. Could you please discuss the possibility of using computer image analysis technique for analysis of different agriculture processes as: mixing of agriculture products, drying, evaluation of quality work of field machinery during crop planting etc.
Thank you for the reviewer’s recommendation. The possibility of using computer imaging analysis technique for the analysis of different agricultural processes is discussed in the revised manuscript from line 106-114. In our opinion, a thorough review on the computer image analysis technique will be more interesting if this topic is investigated in another comprehensive review.
2. Please correct english - some letters are missing.
The language proficiency of the entire manuscript has been carefully revised. Several texts are corrected such as:
Saccharose -> sucrose (Table 6)
Evalution -> Evaluation (Table 7)
Oil palm -> Natural rubber (Line 856)
Last but not least, we would like to take this opportunity to thank all the reviewers for their critical reviews and ingenious thoughts. Without them, we cannot improve the quality of the manuscript.
Your supports and co-operations are highly appreciated and thank you so much in advance.
Reviewer 2 Report
The manuscript provides succinct review of optical sensing technology namely imaging, spectroscopic and spectral imaging for agricultural applications. Authors have very nicely reviewed all the three technologies and provided the theoretical and practical information of the same. I would like the authors to consider the following suggestions to improve the manuscript:
L85- Imaging spectroscopy is not the correct word. You might want to replace it with the imaging technique.
Although you have provided a detailed review of the technologies and given its scope of applications, the manuscript lacks in the review of the application of these technologies in the food and agricultural sectors. Table 5,6 and 7 are very nice to have, but not enough for the review paper. Discussion of research methodologies including instrument, algorithms, samples, and result should be written in text.
Section 4 is impressive. Please include similar review for the food safety inspection and quality control.
Some comparison of published works, their advantages, and shortcomings, future research directions are needed. In the "Possible Challange" section, you have mainly focused on implementing the optical sensors in practice (which is impressive), but have not touched the prevailing research challenges and opportunities.
Author Response
We would like to sincerely thank you very much for taking time, out of your busy schedule to manage the review process. We would equally wish to thank reviewers for their considerations. We have carefully considered all the comments made by reviewers appertaining to the above article and made the following changes to clarify some of his/her concerns and thus improve the overall presentation of the paper, and its quality and clarity.
The details of modifications and our replies are as follows:
Reviewer 2
1. L85-Imaging spectroscopy is not the correct word. You might want to replace it with the imaging technique.
Thank you for the reviewer’s suggestion. We agree that L85-Imaging spectroscopy is not the correct word, and it was replaced with the imaging technique in the revised manuscript.
2. Although you have provided a detailed review of the technologies and given its scope of applications, the manuscript lacks in the review of the application of these technologies in the food and agricultural sectors. Table 5, 6 and 7 are very nice to have, but not enough for the review paper. Discussion of research methodologies including instrument, algorithms, samples, and result should be written in text.
Thank you for the reviewer’s suggestion. Discussion of research methodologies including instrument, algorithms, samples, and result have been written in the revised manuscript for the three techniques:
(a) Imaging – Line 546-583.
(b) Spectroscopy – Line 590-662.
(c) Spectral imaging – Line 673-737.
3. Section 4 is impressive. Please include similar review for the food safety inspection and quality control.
Thank you for the reviewer’s comment. A similar review for the food safety inspection and quality control has been updated in the revised manuscript, Section 4.
4. Some comparison of published works, their advantages, and shortcomings, future research directions are needed. In the "Possible Challenge" section, you have mainly focused on implementing the optical sensors in practice (which is impressive), but have not touched the prevailing research challenges and opportunities.
Thank you for the reviewer’s recommendation. Apart from discussing the implementation of the optical sensors in practice, the prevailing research challenges and opportunities are highlighted in the “Possible Challenge” section from line 902-918 in the revised manuscript.